# Design and Characterization of a Burst Mode 20 Mfps Low Noise CMOS Image Sensor

**DOI:** 10.3390/s23146356

**Published:** 2023-07-13

**Authors:** Xin Yue, Eric R. Fossum

**Affiliations:** Thayer School of Engineering at Dartmouth, Dartmouth College, Hanover, NH 03755, USA

**Keywords:** ultra-high speed, high conversion gain, low noise, image sensor

## Abstract

This paper presents a novel ultra-high speed, high conversion-gain, low noise CMOS image sensor (CIS) based on charge-sweep transfer gates implemented in a standard 180 nm CIS process. Through the optimization of the photodiode geometry and the utilization of charge-sweep transfer gates, the proposed pixels achieve a charge transfer time of less than 10 ns without requiring any process modifications. Moreover, the gate structure significantly reduces the floating diffusion capacitance, resulting in an increased conversion gain of 183 µV/*e*−. This advancement enables the image sensor to achieve the lowest reported noise of 5.1 *e*− rms. To demonstrate the effectiveness of both optimizations, a proof-of-concept CMOS image sensor is designed, taped-out and characterized.

## 1. Introduction

Ultra-high-speed (UHS) image sensors find extensive application in various fields, including medical, scientific, and industrial domains, enabling visualization and understanding of UHS phenomena. Several recent studies [1,2,3,4,5,6,7,8] have successfully achieved frame rates of up to hundreds of millions of frames per second (Mfps) for these specialized sensors. However, these achievements rely heavily on advanced processes, such as 130 nm backside illumination (BSI)-Charge-Coupled Device (CCD) or customized processes tailored specifically for this application. Unfortunately, these advanced/customized processes often come with prohibitively high costs or limited accessibility due to resource constraints. Therefore, there is a pressing need to improve process compatibility. Additionally, the published works commonly encounter relatively high noise due to the inherent trade-off between the design requirement for quick readout speed (favoring smaller capacitance) and the need for lower thermal noise (requiring larger capacitance). Wu et al. [6] reported the lowest state-of-the-art input-referred noise to be 8.4 *e*− rms.

This paper presents a methodology for optimizing the charge transfer time, conversion gain and read noise by introducing the concept of charge-sweep transfer gates. We demonstrate the feasibility of these techniques by implementing them using a standard 180 nm FSI non-stacked process in a 64 × 64-pixel array. Through simulation and characterization, we show that the designed CMOS image sensor has the potential to achieve a frame rate of 20 Mfps and an input-referred noise of 5.1 *e*− rms. Use of these techniques in an advanced 3D-stacked BSI process in the future would lead to further improvements of speed, fill factor, and pixel layout efficiency.

The paper is organized as follows: First, the approach to designing the photodiode and transfer gates is described. Next, the circuitry for in-pixel correlated double sampling (CDS) and the memory array is presented. Then, the sensor noise is calculated and simulated. Finally, characterization results are provided and the discrepancy between simulation and characterization is analyzed.

## 2. Pixel Core Design 

Pixel pitch for high-speed image sensors is larger than for most consumer image sensors in order to increase light gathering for very short integration times. Collections of photoelectrons across larger pixels can take longer than for smaller pixels. It is widely known that there are two main mechanisms for carrier transport in semiconductors [9]: diffusion current, resulting from concentration gradients, and drift current, caused by the presence of an electric field. Electrons can achieve significantly higher velocity in a strong electric field, leading to shorter charge transfer times attributed to drift current compared to diffusion current. Hence, the crucial factor for achieving ultra-high-speed pixels lies in the successful implementation of a strong electric field within the photodiode for fast carrier collection.

In modern pinned photodiodes (PPD) [7,10], the *p+* layer deposited on top of the *n* region plays a crucial role in pinning the surface potential and reducing dark current, depleting the free electrons in the photodiode, and reducing image lag. Analytical studies by Krymski and Feklistov [11] and Park and Uh [12] have pointed out that the maximum electrostatic potential is influenced by the photodiode width, especially for small-size pixels, due to 3D photodiode effects. Figure 1a illustrates a simplified cross-section of a narrow-pinned photodiode. When considering 3D photodiode effects, for a fully depleted *n*-region, the electrostatic potential along the red-dashed line can be plotted in Figure 1b. This potential is described by Equation (1) [13], which provides a simplified relationship between the maximum electrostatic potential (*ψ_max_*) in the photodiode, the elementary charge (*q*), the effective doping concentration of the photodiode (*N_D_*), the doping concentration of the substrate (*N_A_*), and the photodiode’s half-width (*x_n_*).
(1)$$\psi_{max}\approx \frac{{q\cdot N}_{D}\cdot x_{n}^{2}}{2\cdot \varepsilon_{0}\cdot \varepsilon_{r}}\left(1+\frac{N_{D}}{N_{A}}\right)$$

A more advanced 2D model that accounts for the impact of all four PN junctions in a pinned photodiode is available in Liu et al. [14]. However, for quick analysis of the electric field, the simplified 1D model is sufficient. By adjusting the photodiode’s width along its length, it is possible to establish a consistently strong electric field within the photodiode, as shown by Equation (2). In this equation, *x* and *y* denote the coordinates of the photodiode’s envelope, *E* represents the constant electric field, and *C*_0_ stands for a constant.
(2)$$y=-\frac{{q\cdot N}_{D}\cdot x^{2}}{2\cdot {E\cdot \varepsilon }_{0}\cdot \varepsilon_{r}}\left(1+\frac{N_{D}}{N_{A}}\right)+C_{0}$$

Assuming a medium doping concentration at room temperature and an electron mobility of 1500 cm^2^/Vs [15], and considering a charge transfer time that does not exceed 2.5 ns based on system-level calculations, a minimum constant electrical field is calculated to be 370 V/cm. In order to accommodate process variation and the potential occurrence of high dark currents caused by strong electrical fields [16], the electric field within the photodiode along the charge transfer direction ranges from 400 V/cm to 900 V/cm. This results in different photodiode geometric shapes, labeled E400 to E900, as depicted in Figure 2a.

Figure 2b illustrates a conceptual pixel layout that is based on the E900 photodiode depicted in Figure 2a. Through calculations, the photodiode will have a built-in electric field of 900 V/cm. In order to accelerate the simulation process, this simple pixel model only consists of the photodiode (PD), transfer gate (TX), floating diffusion node (FD), reset gate (RST), and reset drain (VDD). The electrostatic potential within the photodiode along the charge transfer path and the charge transfer time, for different photodiode designs, are simulated using Technology Computer-Aided Design (TCAD), and the corresponding results can be found in Figure 3 and Table 1. Simulation results show a stronger electric field than the simple estimation based on Equation (2). This discrepancy may be attributed to the 3D effect of the pinned photodiode. 

In this design, a criterion of 0.5% charge transfer inefficiency (CTI) was selected to ensure satisfactory image lag performance. Considering the system-level requirements, the TX pulse width, including control signal rising and falling time, must be limited to 10 ns. Consequently, only the E700, E800, and E900 designs, as highlighted, meet the criteria. After evaluating the accommodations involving dark current, fill-factor, and process variation, the E800 design was ultimately selected as the high-speed photodiode design in this paper. 

Inspired by the pixel designs presented in Suzuki et al. and Cao et al [1,17] and CCD in CMOS [18,19,20], we propose a high-speed pixel design and introduce the concept of charge-sweep transfer gates, as illustrated in Figure 4a. Each photodiode finger in this pixel is identical to the E800 design shown in Figure 2a, albeit with a rotated angle. Unlike the conventional 4T pixel, our design incorporates TX3, TX2, and TX1 gates to establish a strong electrical field pointing from the tip of the photodiode to the center of the pixel, enabling quick charge transfer. The operational timing of this pixel is depicted in Figure 4b. During the beginning of the charge transfer, all three gates, TX1, TX2, and TX3, are simultaneously turned on. Notably, the voltage of TX1 is higher than that of TX2, and the voltage of TX2 is higher than that of TX3. Upon completion of the charge transfer, the TX3 gate is turned off first, followed by TX2, and finally TX1, sequentially sweeping electrons from TX3 to FD. 

In order to accurately simulate the device, the proposed pixel is also modeled in TCAD, as depicted in Figure 5. For the sake of clarity, the silicon dioxide (SiO_2_) layer that fully covers the entire pixel is not displayed, but TX1, TX2, TX3, RST, SF, SEL, and their respective metal connections are visible. Figure 6 illustrates electrostatic potential plots along the charge transfer path in one PD under various TX voltages on the red-dashed cut-plane. The plot clearly shows the presence of a small potential barrier (~0.1 V) between two adjacent gates due to the gap between gates. By controlling the falling edge slew rate of the TX gates, it is possible to eliminate the potential barrier during the falling transition of the TX gates. This process establishes a monotonically increasing electrostatic potential profile and generates a strong electrical field that sweeps previously stored electrons to the next gate, ultimately enabling a full charge transfer.

A TCAD transient simulation was conducted to verify the complete charge transfer of the proposed design. The results are illustrated in Figure 7, where eTotal1 and eTotal2 represent the total number of electrons in the upper and lower photodiodes, respectively. At the start of the charge transfer process, there are 111 and 92 electrons in the two photodiodes, respectively. Within 10 ns, both eTotal1 and eTotal2 rapidly decrease to less than 1 electron, indicating successful full charge transfer. Following the charge transfer operation, the FD voltage changes from 2.366145 V to 2.338567 V, resulting in a Conversion Gain (CG) of 136 μV/*e*−.

As mentioned previously, high-speed CMOS image sensors are prone to relatively high noise due to the trade-off between design requirements for fast readout speed and lower thermal noise. In order to tackle this issue, a passive correlated double sampling (CDS) amplifier was proposed by Wu et al. [6] to reduce input-referred noise. However, the gain of the passive CDS amplifier relies on the capacitance ratio between the NMOS capacitor in depletion mode and inversion mode, which is dependent on both the process and voltage. Consequently, this introduces an unavoidable non-linearity to the entire image sensor, approximately at a level of 3% [6]. Furthermore, the settling of the amplified voltage imposes limitations on the frame rate. 

Therefore, in this design, the image sensor’s CG was proposed to be improved to minimize input-referred noise. Figure 8a illustrates the schematic of the proposed pixel, highlighting the major capacitance contribution, while Figure 8b displays the capacitance distribution at FD of the pixel, with the dominant factors being the FD-to-ground capacitance (C_fd_gnd_) and the FD-to-Source Follower Drain capacitor (C_sfd_fd_), leaving potential for further optimization.

Figure 9a illustrates a 3D TCAD model showing the default minimal buried channel NMOS in this process, along with its cross-section. It is evident that the effective channel length is significantly shorter than the gate length due to the diffusion of *n* dopants in the source and drain, impeding the reduction of the transistor gate length. Directly reducing the gate length beyond the design rule limit can potentially lead to an increased leakage current between the drain and source, primarily due to drain-induced barrier lowering (DIBL). Moreover, this reduction may induce other short channel effects or even cause a direct short circuit. Based on the ideas presented in Kusuhara et al. and Seo et al. [21,22], we propose the elimination of the lightly-doped-drain (LDD) at the drain side, enabling further transistor shrinkage and reduction of FD capacitance. Figure 9b illustrates this proposed modification, which involves only mask changes.

To determine the optimal NMOS transistor design, we explore the gap distance ranging from 0 µm to 0.3 µm. Both the proposed buried channel NMOS transistor (L = 0.3 µm, gap = 0 µm~0.3 µm) and the default transistor (L = 0.6 µm, gap = 0 µm) are configured as a source follower biased with an ideal DC current sink. The gate voltage was swept from 1.5 V to 2.5 V and the corresponding results are plotted in Figure 10. It is clear that designs with L = 0.3 µm, Gap = 0 µm; L = 0.3 µm, Gap = 0.1 µm; and L = 0.6 µm, Gap = 0 µm demonstrate superior linearity compared to the other designs.

As discussed earlier, eliminating the LDD region results in a notable decrease in the overlap capacitance between the gate and drain. Table 2 presents the C_gd_ values along with other alternating current (AC) parameters for different design configurations.

Taking into account the Miller effect, the design with L = 0.3 and Gap = 0.1 achieves the smallest lumped capacitance at FD. Consequently, this design will be employed in the high CG version of the proposed pixel. 

Concerns may arise regarding the increased flicker noise caused by smaller gate geometries. As mentioned in Boukhayma et al. [23], the smaller SF gate enhances the conversion gain and reduces input-referred noise, particularly when a fast CDS circuit is utilized. Another concern might be the potential occurrence of the hot electron effect in the absence of lightly doped drain (LDD) regions. However, as shown in Figure 9b, a “lightly doped” region still exists on the drain side due to the diffusion effect, and being physically separated from the channel. Therefore, in terms of electrical field, the proposed new design exhibits a lower peak electric field compared to the default design under the same bias conditions. Consequently, the proposed new design presents a lower probability of high-energy collisions than the default design.

A TCAD transient simulation is conducted again to validate the efficiency of the proposed source follower. The transient simulation demonstrates a significant increase in CG, from 136 μV/*e−* to 178 μV/*e−*.This result closely aligns with the findings from the previous AC simulation. 

To distinguish between these two pixels throughout the remainder of this paper, the pixel with the default source follower will be referred to as the baseline pixel, while the pixel employing the proposed source follower will be referred to as the HCG (high conversion gain) pixel.

## 3. In-Pixel CDS and Memory Bank

In most CMOS image sensors, input-referred noise is primarily dominated by in-pixel SF thermal and flicker noise. To mitigate low-frequency noise originating from the pixel source follower, pixel reset kTC noise, and fixed-pattern noise (FPN), we have implemented the correlated-double-sampling (CDS) circuit [24] shown in Figure 11 for this project. To minimize voltage gain attenuation in the signal chain, we have positioned the sample-and-hold capacitor (C_SH_) at the output of the first-stage source-follower, contrary to its placement at the input of the second-stage source-follower, as described in Miyauchi et al. [25]. This arrangement effectively reduces voltage attenuation in the signal chain to C_CDS_/(C_CDS_ + C_P_), where C_CDS_ represents the AC CDS capacitor and C_P_ represents the parasitic capacitor. Its operation timing is shown in Figure 4b.

According to the design requirements, each pixel requires at least 100 sample-and-hold capacitors. To increase the capacitance density, a custom Metal-1 (M1) Metal-Oxide-Metal (MOM) capacitor is placed on the top of the 1.8 V thin poly gate of the NMOS capacitor. Moreover, a Metal-2 (M2) layer is utilized as a shielding layer positioned above the M1 MOM capacitor, as depicted in Figure 12. To protect the 1.8 V thin gate devices in a 3.3 V environment, the V_RST_ voltage is isolated from VDD_pix_ and can be independently adjusted. Typically, the V_RST_ voltage is set to 1.8 + V_GS_SF2_ to ensure that the maximum output voltage of SF2 remains below 1.8 V.

With this design, a total of 108 units of sample and hold capacitors, each having a capacitance of 78 fF, can be accommodated within a 52.8 µm pixel in the final layout. To reduce the capacitance loading of source follower SF2 and minimize possible charge corruption, a capacitor bank with a hierarchical switches network and individual controls is used. The schematic of the final pixel is illustrated in Figure 13.

## 4. Sensor Noise Estimation 

Despite the high pixel conversion gain designed for this image sensor, it is still crucial to meticulously analyze the noise of the signal chain. The primary noise sources within this sensor have been identified and highlighted in red in Figure 13.

Contributor 1 represents the flicker noise and thermal noise originating from the in-pixel 1st stage source follower (SF1). A thorough measurement and validation of the flicker noise model were performed by Deng and Fossum [26] for Taiwan Semiconductor Manufacturing Company (TSMC) processes, revealing that the Hooge mobility fluctuation model matches the experimental measurements best. Even though a different process is used in this design, the model can still be used for quick calculation. Equation (3) displays the normalized power spectrum density (PSD) of the flicker noise, as presented in Deng and Fossum [26], where *α_H_* stands for Hooge’s parameter and *C_ox_* refers to the oxide capacitance per unit area. Equation (4) describes the classical thermal noise spectrum of SFs, where *g_m_* denotes the transconductance of the source follower, *r* is the excess noise factor, and *k* is Boltzmann’s constant.
(3)$$\frac{s_{I_{b\_flicker}\left(f\right)}}{I_{b}^{2}}=\frac{\alpha_{H}}{f}\times \frac{2q}{C_{ox}WL\times (V_{GS}-V_{th})}$$
(4)$$s_{I_{b\_thermal}\left(f\right)}=4kTrg_{m}$$

Given the limited bandwidth of SFs, the transfer function of the in-pixel 1st SF can be treated as a low-pass filter. This behavior is depicted by Equation (5), where *f_c_* denotes the cutoff frequency.
(5)$$H_{LP1\left(f\right)}=\sqrt{\frac{1}{1+{\left(f∕f_{c}\right)}^{2}}}$$

Therefore, the total noise voltage produced by contributor 1 at its output can be obtained by consolidating Equations (3)–(5), as shown in Equation (6). In this equation, *A_SF_* represents the gain of the source follower and Δt stands for the time difference between pixel reset votlage sampling and pixel signal votlage sampling in CDS operations.
(6)$$V_{n1}^{2}=\int_{0}^{\infty }\frac{s_{I_{b1\_thermal}\left(f\right)}+s_{I_{b1\_flicker}\left(f\right)}}{g_{m1}^{2}}\times A_{SF1}^{2}\times H_{LP1\left(f\right)}^{2}\times ({2\times \sin\left(\pi f\Delta t\right))}^{2}$$

Contributor 2 denotes the kTC noise arising from the Brownian motion of carriers within the RST2 switch. As per classical theory, the kTC voltage noise is determined by Equation (7), where C denotes the capacitance of the CDS capacitor.
(7)$$V_{n2}^{2}=\frac{kT}{C}$$

Contributor 3 represents the flicker noise and thermal noise generated by the in-pixel 2nd-stage source follower (*SF*2). It is worth noting that a relatively larger transistor size is required to reduce the flicker noise, given the absence of CDS noise cancellation. Its overall noise contribution, as depicted in Equation (8), is similar to that of contributor 1.
(8)$$V_{n3}^{2}=\int_{0}^{\infty }\frac{s_{I_{b2\_thermal}\left(f\right)}+s_{I_{b2\_flicker}\left(f\right)}}{g_{m2}^{2}}\times A_{SF2}^{2}\times H_{LP2\left(f\right)}^{2}$$

Contributor 4 represents the kTC noise from the sample-and-hold switch. Similar to contributor 2, the voltage noise can be obtained by Equation (9), where *C_SH_* stands for the capacitance of the sample-and-hold capacitor.
(9)$$V_{n4}^{2}=\frac{kT}{C_{SH}}$$

Contributor 5 represents the kTC noise from the RST3 switch, which will be attenuated by the sample-and-hold capacitor. The total equivalent noise can be calculated by Equation (10).
(10)$$V_{n5}^{2}=\frac{kT}{C_{para2}}\times {\left(\frac{C_{para2}}{C_{para2}+C_{SH}}\right)}^{2}$$

Contributor 6 denotes the flicker noise and thermal noise that arise from the in-pixel 3rd-stage source follower (*SF*3). Similar to contributor 3, the cumulative noise at the output can be determined using Equation (11).
(11)$$V_{n6}^{2}=\int_{0}^{\infty }\frac{s_{I_{b3\_thermal}\left(f\right)}+s_{I_{b3\_flicker}\left(f\right)}}{g_{m3}^{2}}\times A_{SF3}^{2}\times H_{LP3\left(f\right)}^{2}$$

Contributor 7 denotes the thermal noise originating from the column output buffer (not shown in the figure), which is a simple 5 transistors operational transconductance amplifier (OTA). Considering the comparatively larger size of the input pair, its flicker noise component is negligible. The overall output noise of the buffer can be determined by Equation (12).
(12)$$V_{n7}^{2}=2\left(4kTrg_{m1}+4kTrg_{m2}\right)\times R_{out}^{2}\times {BW}^{2}$$

Therefore, the total output voltage noise of the whole image sensor can be estimated by Equation (13). It is crucial to note that, for the sake of quick calculation, any additional bandwidth limiting caused by subsequent stages has been disregarded in this simple calculation. The variables used in this equation include *A_CDS_*, which represents the attenuation introduced by the CDS capacitor, *A_SF_*_2_, the low frequency gain of the 2nd-stage source follower, *A_SH_*, the attenuation introduced by the sample-and-hold capacitor and *C_para_*_2_, *A_SF_*_3_, the low frequency gain of the 3rd-stage source follower, and *A_UGB_*, the low frequency gain of the column output buffer.
(13)$$\begin{array}{cc} V_{n_{output}}^{2}= & V_{n1}^{2}\times A_{CDS}^{2}\times A_{SF2}^{2}\times A_{SH}^{2}\times A_{SF3}^{2}\times A_{UGB}^{2}\\ & \;\;\;+V_{n2}^{2}\times A_{SF2}^{2}\times A_{SH}^{2}\times A_{SF3}^{2}\times A_{UGB}^{2}+V_{n3}^{2}\times A_{SH}^{2}\times A_{SF3}^{2}\times A_{UGB}^{2}\\ & \;\;\;+V_{n4}^{2}\times A_{SF3}^{2}\times A_{UGB}^{2}+V_{n5}^{2}\times A_{SF3}^{2}\times A_{UGB}^{2}+V_{n6}^{2}\times A_{UGB}^{2}\\ & \;\;\;+V_{n7}^{2} \end{array}$$

The Spectre AC noise simulation is used for quick noise estimation. Table 3 shows the simulated values for *V_n_*_1_ to *V_n_*_7_ and the gain of each stage. According to the simulation results and Equation (13), the estimated output-referred noise at the pad is 414 µV root-mean-square (rms), while the input-referred noise at the FD node is 5.8 *e−*, assuming that the CG is 136 µV/*e−* for a baseline pixel.

The noise of the high-conversion gain (HCG) version is also estimated. Although a Simulation Program with Integrated Circuit Emphasis (SPICE) model for the proposed source follower is unavailable, it is still possible to estimate the flicker noise and thermal contribution based on Equations (3) and (4). Table 4 presents the total noise values for the HCG pixel. According to the simulation results, the output-referred noise at the pad is 418 µV rms, while the input-referred noise at the FD node is 4.6 *e−*, assuming that the CG is 178 µV/*e*− for a baseline pixel.

## 5. Characterization

The sensor was fabricated in a standard 180 nm PPD process. Figure 14 displays the microscopic image of the designed sensor and its prototype testing system. In this version, three types of pixels were taped out. Baseline pixels (64 pix × 32 pix) are positioned on the left side of the pixel array, HCG pixels (32 pix × 32 pix) are located on the upper right of the pixel array, and test pixels (32 pix × 32 pix) are situated on the bottom right of the pixel array, as indicated in Figure 14a.

Although TCAD simulations have demonstrated the sensor’s capability to operate at a minimum of 20 Mfps [27], the current prototype camera system faces limitations due to the hardware capabilities of the Field Programmable Gate Arrays (FPGA), prototype Printed circuit board (PCB), and chip carrier. These constraints impose a maximum reliable operation of 15.6 Mfps. Moreover, the parasitic inductances of the chip carrier introduce substantial ringing on the power supply V_RST_ during pixel reset operations and prolong the CDS time. As a result, to achieve optimal noise performance, the imager noise is currently measured at 4 Mfps. 

It is important to mention that conventional image sensors typically measure noise in completely dark conditions to minimize the impact of photon shot noise. However, in the case of this ultra-high-speed image sensor, the contribution of both photon shot noise and dark current must be taken into account. Therefore, during noise measurement, the TX1 voltage is kept low (“off”). The measured data reveals that the output-referred noise at the pad is 10.9 DN for baseline pixels, corresponding to 415 µV, and 12.0 DN for HCG pixels, corresponding to 457 µV. Here, 1 DN is equal to 38 µV. These voltage noise values closely align with the initial estimations reported in Table 3 and Table 4. 

For photon transfer curve (PTC) measurement, to mitigate the impact of dark current, which can result in inaccurate CG measurements, a powerful light source is used, and the integration times are deliberately kept relatively low. The measured result yields PTC slopes of 0.41 and 0.50 for the baseline pixel and HCG pixels, respectively, in the region dominated by photon shot noise. These values are in close proximity to the ideal value of 0.5. Consequently, it is reliable to determine the pixel CG under this setup. After being adjusted by the signal chain gain of 0.485 V/V, the calculated baseline pixel CG is found to be 98 µV/*e*− and the HCG pixel CG is found to be 183 µV/*e*−. Therefore, the sensor’s input-referred noise was determined to be 8.7 *e−* for the baseline pixel and 5.1 *e*− for the HCG pixel.

Based on the same setup, we measured the full well capacity (FWC) of two pixels as well. The measured data shows that baseline pixel output reaches saturation at approximately 8500 DN, equivalent to 6800 *e*−, and the HCG pixel output saturates at around 11,600 DN, corresponding to 5000 *e*−.

At 300 K, the baseline pixel shows a dark electron rate at 0.66 *e−*/ns/pixel, whereas the HCG pixel exhibits 0.46 *e−*/ns/pixel. The measured results exceed the expected values and current evidence suggests a higher than anticipated trap state density in the silicon bulk. According to the classical Shockley-Read-Hall recombination (SRH) theory [28,29,30], reducing the temperature can decrease the dark current rate. At 256 K, both the baseline pixel and the HCG pixel exhibit a dark electron rate of less than 0.01 *e−*/ns/pixel. For the intended application, where the frame rate is 20 Mfps, the maximum theoretically possible integration time is 50 ns. Consequently, the maximum number of dark electrons is only 0.5, significantly smaller than the sensor’s noise floor. 

The sensor is specifically designed to capture ultra-high-speed events occurring within microseconds while maintaining minimal image lag. Currently limited by the speed of the testing light source, the sensor was slowed down to 142 Kfps to perform the lag test, with the pixel TX3 pulse duration fixed at 10 ns. Based on the measurement results, the baseline pixel exhibits a negligible lag of 0.03%. On the other hand, the HCG pixel shows a lag of approximately 3% due to overflow at the floating diffusion node. However, by adjusting the TX gate’s negative off voltage, it was possible to mitigate the lag.

Figure 15a illustrates the test setup, with a camera lens mounted on the PCB to demonstrate a video capturing the falling edge of a focused LED array, where LEDs were on for the first six frames and stayed off for the remaining frames. Figure 15b shows the image of an LED array and its hand-made driver circuit. Figure 15c exhibits the captured video, comprised of 108 frames. It is important to note that the LED array has a limited light intensity. Therefore, to collect enough photons per frame, the image sensor needs to slow down to 400 Kfps for this test. The captured images show non-uniformity, which is caused by LED non-uniformity. 

## 6. Conclusions

This paper presents the design and characterization of an ultra-high-speed burst-mode low-noise CMOS image sensor. The most challenging aspect of the design is implementing it in a standard 180 nm PPD process. Overcoming this challenge involves achieving full charge transfer within 10 ns and minimizing floating diffusion capacitance without making any process modifications. To do so, the concept of process-independent charge-sweep transfer gate was invented and optimized. The simulation and measurement results demonstrate an acceptable match in terms of noise, pixel conversion gain, and charge transfer time. Table 5 summarizes the sensor characterization results and Table 6 shows a comparison of this work to related state-of-the-art sensors.

Demonstrated performance of the sensor, while consistent with simulation and modeling, was limited by both the use of an off-the-shelf package and by the brightness of the test system modulated light sources. An improved packaging and test system environment can be explored in the future.

## Figures and Tables

**Figure 1 sensors-23-06356-f001:**
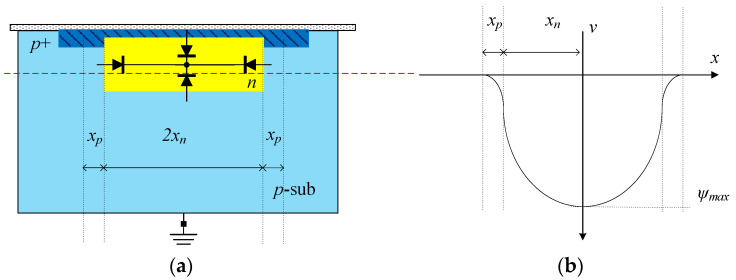
(**a**) Simplified cross-section of a narrow PPD. (**b**) Electrostatic potential along the red-dashed line.

**Figure 2 sensors-23-06356-f002:**
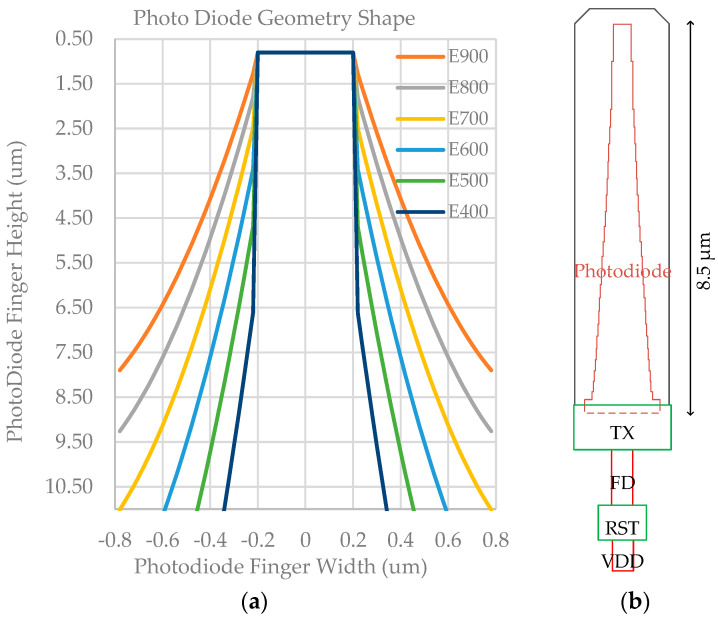
(**a**) Photodiode finger shapes with different electric fields. (**b**) Conceptual layout of a pixel with a built-in 900 V/cm electric filed.

**Figure 3 sensors-23-06356-f003:**
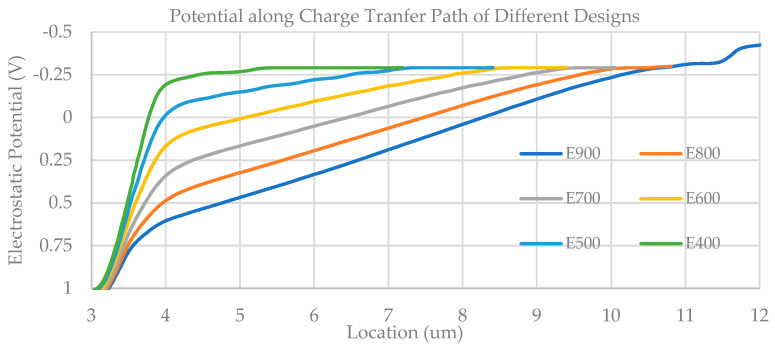
Electrostatic potential along the charge transfer path for various photodiode designs.

**Figure 4 sensors-23-06356-f004:**
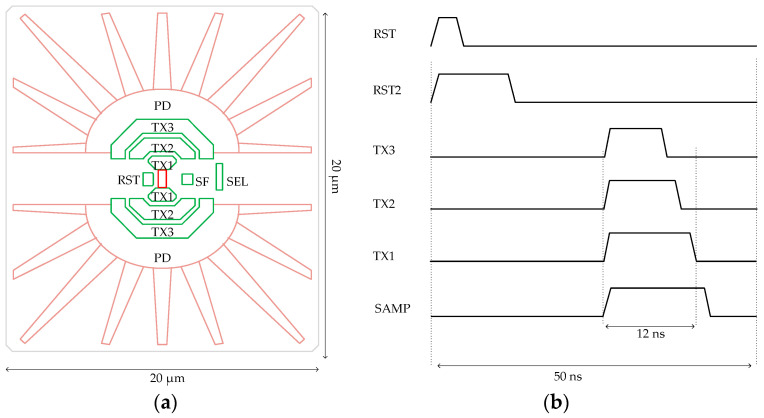
(**a**) Conceptual layout of the proposed high-speed pixel. (**b**) Operation timing of a charge-sweep-gate based pixel.

**Figure 5 sensors-23-06356-f005:**
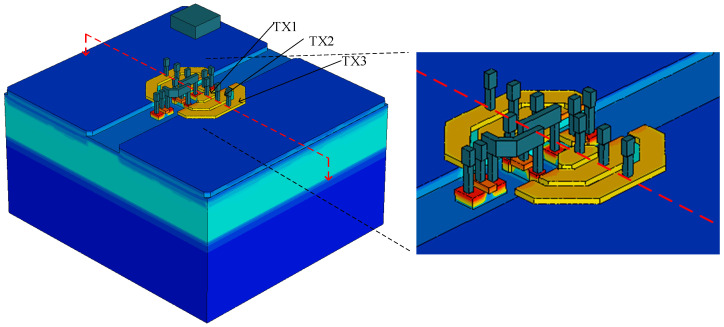
TCAD 3D model of the proposed pixel with expanded view.

**Figure 6 sensors-23-06356-f006:**
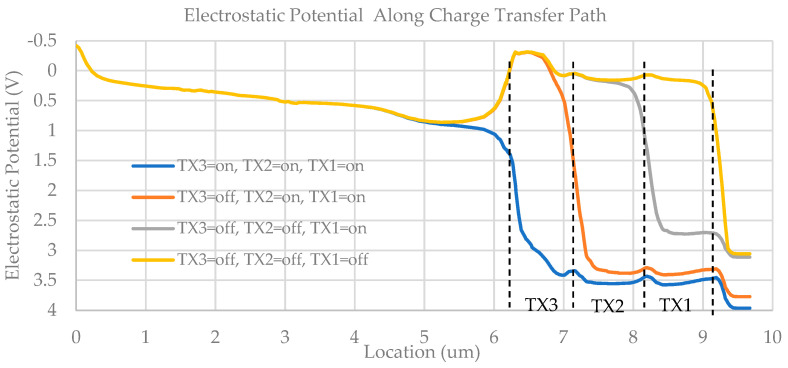
Electrostatic potential plots along the charge transfer path during charge transfer.

**Figure 7 sensors-23-06356-f007:**
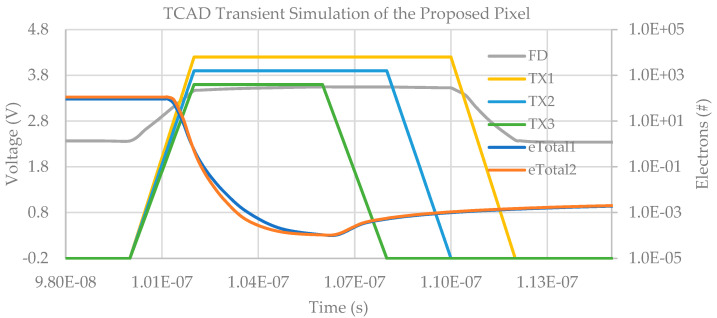
TCAD transient simulation result of the proposed pixel.

**Figure 8 sensors-23-06356-f008:**
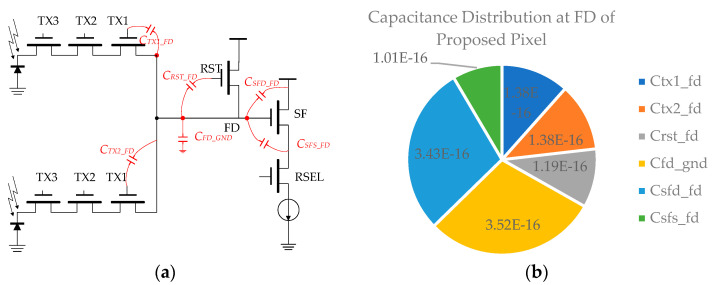
(**a**) Schematic of the proposed high-speed pixel. (**b**) Capacitance distribution at FD node.

**Figure 9 sensors-23-06356-f009:**
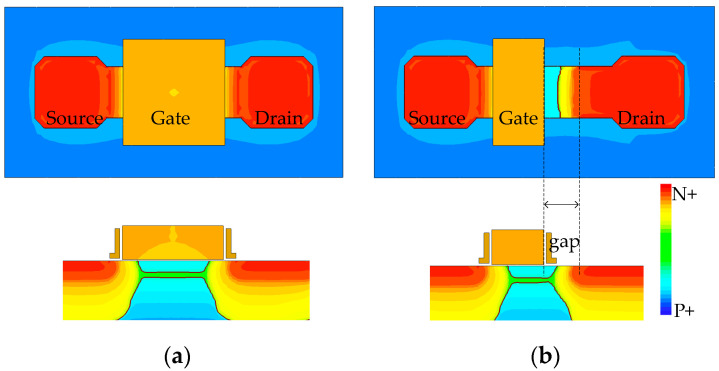
(**a**) 3D TCAD model of the default buried channel NMOS and its cross-section. (**b**) 3D TCAD model of the proposed buried channel NMOS and its cross-section.

**Figure 10 sensors-23-06356-f010:**
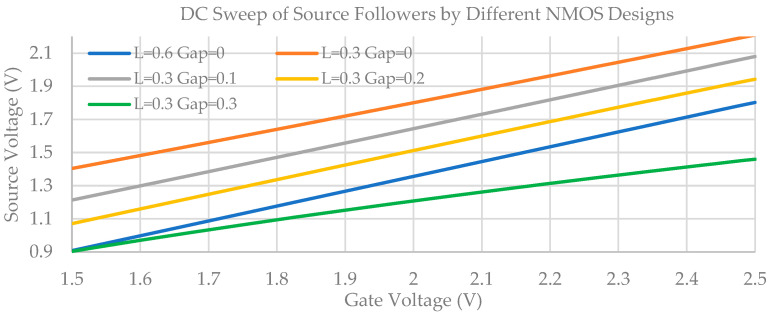
DC sweep simulation results of buried channel NMOSs.

**Figure 11 sensors-23-06356-f011:**
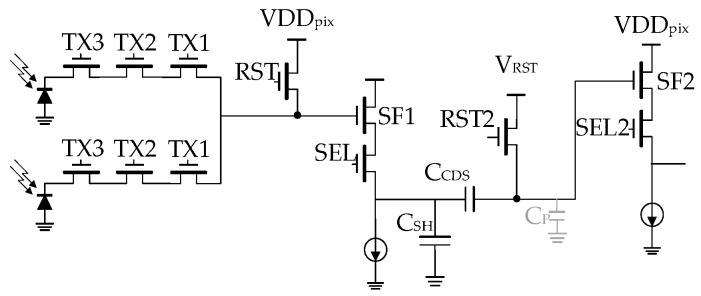
In-pixel CDS circuit.

**Figure 12 sensors-23-06356-f012:**
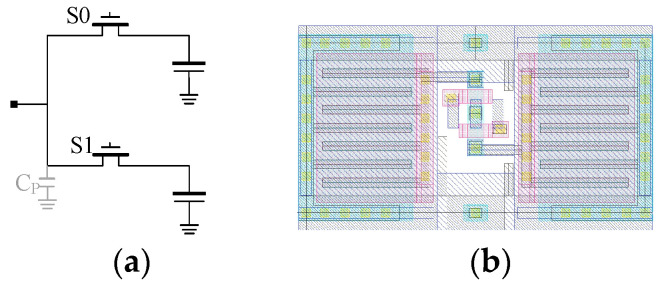
(**a**) schematic and (**b**) layout of in-pixel storage unit.

**Figure 13 sensors-23-06356-f013:**
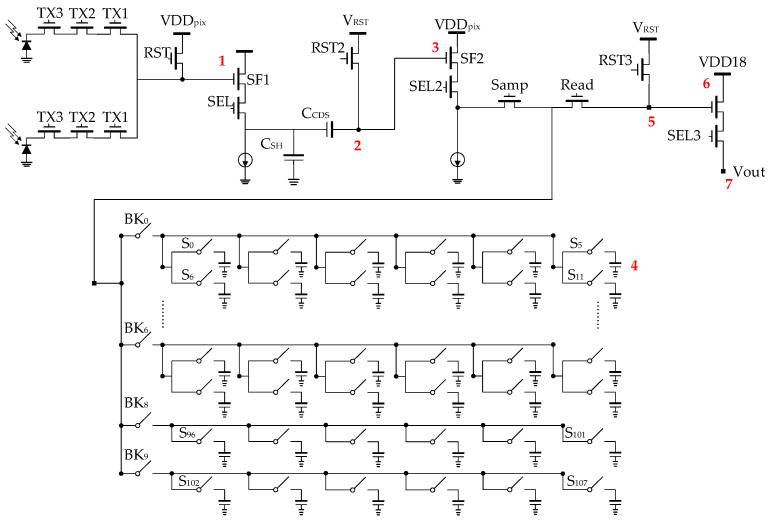
Complete pixel schematic with 108 sample-and-hold capacitors.

**Figure 14 sensors-23-06356-f014:**
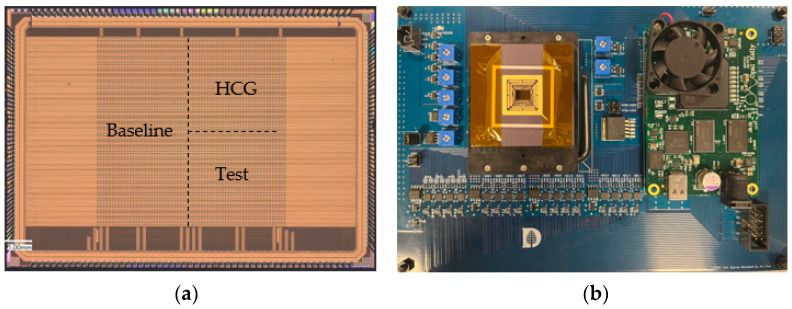
(**a**) The microscopic image of the designed image sensor chip. (**b**) The protype test system.

**Figure 15 sensors-23-06356-f015:**
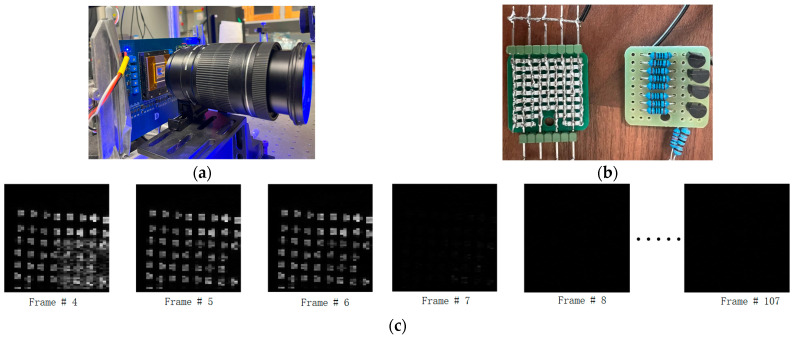
(**a**) Test setup for video capturing. (**b**) LED array and its driver circuit. (**c**) Images captured by the sensor at 400 Kfps. LEDS are on for frames 1–6 and off for remaining frames.

**Table 1 sensors-23-06356-t001:** Charge Transfer Time Simulation Results of Different Designs.

CTI	10%	1%	0.5%	0.1%	Unit
E400	10.2	41.2	51.3	75.2	ns
E500	0.6	17.4	25.8	47.0	ns
E600	0.6	5.0	11.0	28.3	ns
E700	0.7	1.5	**5.1**	19.3	ns
E800	0.8	1.1	**3.7**	15.5	ns
E900	0.9	1.3	**3.6**	13.6	ns

**Table 2 sensors-23-06356-t002:** AC Performance of Different SF Designs.

SF Length (µm)	Gap Dist. (µm)	Gain@Vg = 2.5 V (V/V)	Gain@Vg = 1.5 V (V/V)	C_gs_ (fF)	C_gd_ (fF)
0.3	0.0	0.82	0.77	0.52	0.22
0.3	0.1	0.87	0.85	0.57	0.17
0.3	0.2	0.83	0.88	0.66	0.15
0.3	0.3	0.47	0.68	0.68	0.12
0.6	0.0	0.89	0.89	0.79	0.26

**Table 3 sensors-23-06356-t003:** Sensor noise estimation based on a baseline pixel.

Noise Source	Cap Size (pF)	Stage Noise (µV)	Stage Gain (V/V)	Noise Contribution (µV^2^)	Noise Percentage (%)	Total Noise (µV)
1. 1st-Stg SF		374	0.81	213,193	33	
2. Rst2 kTC	0.080	233	0.96	82,819	13	
3. 2nd-Stg SF		191	0.90	74,485	12	
4. S/H kTC	0.078	236	0.78	113,788	18	
5. Rst3 kTC		169	1.00	95,849	15	
6. 3rd-Stg SF		095	0.89	38,237	06	
7. Out Buffer		082	1.00	28,488	04	
Noise @ FD						804
Noise @ Pad						414

**Table 4 sensors-23-06356-t004:** Sensor noise estimation based on HCG pixel.

Noise Source	Cap Size (pF)	Stage Noise (µV)	Stage Gain (V/V)	Noise Contribution (µV^2^)	Noise Percentage (%)	Total Noise (µV)
1. 1st-Stg SF		385	0.81	225,918	34	
2. Rst2 kTC	0.080	233	0.96	82,819	13	
3. 2nd-Stg SF		191	0.90	74,485	11	
4. S/H kTC	0.078	236	0.78	113,788	17	
5. Rst3 kTC		169	1.00	95,849	15	
6. 3rd-Stg SF		095	0.89	38,237	06	
7. Out Buffer		082	1.00	28,488	04	
Noise @ FD						812
Noise @ Pad						418

**Table 5 sensors-23-06356-t005:** Sensor Characteristics Summary.

Sensor Characteristics Summary					
Process	180 nm standard PPD CIS				Unit
Pixel Pitch	52.8 × 52.8				µm × µm
Pixel Fill Factor	9.7				%
Pixel Array Size	64 × 64				pix × pix
Recording Length	108				frames
Pixel Variant	Baseline		HCG		
Measurement/Simulation	Mesa.	Sim.	Meas.	Sim.	
Charge Transfer Time	≤10	≤10	≤10	≤10	ns
Conversion Gain	98	136	183	178	µV/*e−*
Output-Referred Noise	415	414	457	418	µV
Input-Referred Noise	8.7	5.8	5.1	4.6	*e−*
Image Lag	≤0.1	≤0.5	≤3	≤0.1	%
FWC	6.0	7.0	5.0	5.6	K*e*−
Low Light Linearity	±0.5	±0.2	±0.5	±0.2	%
Dark Current (300 K)	6.9 × 10^−1^	1.6 × 10^−4^	4.6 × 10^−1^	1.6 × 10^−4^	*e−*/ns/pixel
Dark Current (256 K)	1.0 × 10^−2^	N/A	9.2 × 10^−3^	N/A	*e−*/ns/pixel

**Table 6 sensors-23-06356-t006:** Performance Comparison with recently published high speed image sensors.

Ref.	Node (nm)	Process Modi?	Array (H × V)	Pitch (µm)	CG (µV/*e−*)	FWC (K*e*−)	Frame Rate (Mfps)	Record Length	Noise (*e−*)
[1]	180 FSI	Yes	50 × 108	35	99	11	100	368	N/R
[2]	180 FSI	Yes	400 × 256	32	74	N/R	10	128	N/R
[3]	180 FSI	Yes	96 × 128	32	112	10	10	480	N/R
[4]	130 BSI	Yes	32 × 32	72.5	N/R	N/R	25	1220	N/R
[5]	130 CCD	Yes	512 × 575	12.7	N/R	7	100	5	N/R
[6]	130 BSI	N/R	32 × 84	30	105	6	20	108	8.4
[7]	110 FSI	Yes	212 × 188	22.4	32	33	303	12	85
[8]	110 FSI	Yes	320 × 324	11.2	N/R	10	200	15	>167
[31]	90 + 40	N/R	20 × 20	50	7.3	137	5	52	>81
This work	180 FSI	No	64 × 64	52.8	183	5	20	108	5.1

## Data Availability

Data available on request.

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
