# Peer review of "Design and Characterization of a Burst Mode 20 Mfps Low Noise CMOS Image Sensor"

_sensors, 2023, doi:10.3390/s23146356_

Round 1

Reviewer 1 Report

See attached file.

Author Response

Please see the attachment for our point-to-point response to the reviewer's comment.

Reviewer 2 Report

This paper describes the design of an ultra-high speed CMOS image sensor (CIS) that implements charge-sweep transfer gates. The design is fabricated in a standard 180nm CIS process and part of the characterization is described.
The authors did a great job and wrote a very interesting paper going through the design, characterization, and comparison of the simulation results with the measured data.
Here are some minor comments which might improve the paper.
- page. 10, eq. 6, could you add something on the  H^2_CDS(f);
- eq. 7, 9,10, please don't write K uppercase (which means Kelvin) considering other equations (4 and 12) in which k is lowercase.
- moreover, k is kb the Boltzmann's constant and not a "process-dependent parameter" as written in line 254. Please amend it.
- line 36, "Through simulation and characterization, we show that the designed CMOS image sensor can achieve a frame rate of 20 Mfps". The characterization doesn't show this (lines 325-330). I would change this sentence.

Author Response

Please see attached document for our point-to-point response to the reviewer's comments.

Reviewer 3 Report

1. What are the main applications of proposed pixel of the paper due to its large pixel size, ultra high speed, and low resolution?

2. Compared to Table 6, the pixel performance of this paper has little advantage compared to existing technologies, especially in terms of speed. So what are the technical advantages of the proposed design.

Author Response

Please see the attached document for our point-to-point response to the reviewer's comments.
